# Nutrition in Chronic Kidney Disease—The Role of Proteins and Specific Diets

**DOI:** 10.3390/nu13030956

**Published:** 2021-03-16

**Authors:** Mugurel Apetrii, Daniel Timofte, Luminita Voroneanu, Adrian Covic

**Affiliations:** 1Department of Nephrology, University of Medicine and Pharmacy “Grigore T. Popa”, 700115 Iasi, Romania; mugurelu_1980@yahoo.com (M.A.); lumivoro@yahoo.com (L.V.); accovic@gmail.com (A.C.); 2Surgical Department I, University of Medicine and Pharmacy “Grigore T. Popa”, 700115 Iasi, Romania

**Keywords:** nutrition, chronic kidney disease, low protein diet, healthy dietary patterns

## Abstract

Chronic kidney disease (CKD) is a global public health burden, needing comprehensive management for preventing and delaying the progression to advanced CKD. The role of nutritional therapy as a strategy to slow CKD progression and uremia has been recommended for more than a century. Although a consistent body of evidence suggest a benefit of protein restriction therapy, patients’ adherence and compliance have to be considered when prescribing nutritional therapy in advanced CKD patients. Therefore, these prescriptions need to be individualized since some patients may prefer to enjoy their food without restriction, despite knowing the potential importance of dietary therapy in reducing uremic manifestations, maintaining protein-energy status.

## 1. Introduction

Proteins are highly complex, larges sized molecules, that are present in all living organisms being of great nutritional value and involved in the many chemical processes essential for life. Even the term “protein” suggests their importance, as this word is derived from the Greek prōteios, meaning “holding first place.” However, a high protein diet of animal origin, usually associated with Western dietary practices, is consistently associated with a small to moderate increased risk of premature mortality and deleterious effects for numerous chronic diseases, including CKD [1]. 

A high consumption of proteins could be detrimental to kidney function through several mechanisms. First, it may induce vasodilation of afferent renal arterioles, glomerular hypertension, and hyperfiltration, which together accelerate the progression of pre-existing CKD. Second, increased consumption of red and processed meat is associated with an increased blood pressure (caused by the concomitant high intake of sodium chloride), metabolic acidosis, mitochondrial oxidative stress (triggered by saturated fats), DNA damage (caused by N-nitroso compounds), and increased accumulation of the end-products of protein catabolism (such as p-cresyl sulfate, indoxyl sulfate, and trimethyl aminoxide) [2,3]. Therefore, CKD patients are advised by their nephrologists to restrict their protein intake with the main goal of reducing the accumulation of such molecules reducing thus the onset and the severity of uremic symptoms [2]. 

In contrast to meat-based diet, a diet rich in protein from plant sources may be beneficial, preventing heart disease and hypertension as well as delaying the progression of kidney disease. However, the optimal diet for CKD patients remains controversial, depending upon the estimated glomerular filtration rate (eGFR), type of kidney disease (i.e., proteinuric or nonproteinuric), and the presence of other comorbidities such as diabetes, hypertension, or heart failure. 

In this narrative review, we present a summary of the available published data on the impact of low-protein diets and dietary patterns on chronic kidney disease-related outcomes. A literature search was performed as appropriate for narrative reviews, including electronic databases of PubMed, Cochrane Library, and Google Scholar using a combination of the MESH terms: “CKD”, “nutrition”, and low protein diet”, “keto-analogues”, “Mediterranean diet, DASH diet”. All the articles published in the medical literature relevant to the queries were selected and evaluated for relevance to each of the domains selected for review.

## 2. Protein Restriction Alone

Low-protein diet (LPD) is a long-standing recommendation for CKD management, based on its potential protective effect on renal hemodynamic. Moreover, limiting protein intake from animal sources and shifting toward a vegetable protein sources is also associated with favourable effects, including reduction of uremic toxins and correction of metabolic acidosis, in addition to reduced phosphorus load with better control of metabolic bone disorder. These diets should be progressively installed to allow careful dietary monitoring and adequate adherence. Although such diets are not associated with wasting in carefully monitored research studies, on a routine basis, attention should be focused on energy intake, which may decrease over time and induce weight loss and wasting.

Even if the results of studies that have examined protein restriction alone were variable, the balance of evidence suggests a benefit of CKD progression of moderate dietary protein restriction (0.6–0.8 g/kg/day) (see Table 1). The largest trial to date, the Modification of Diet in Renal Disease (MDRD) study, analyzed a large cohort of CKD nondiabetic patients with a GFR of 25 to 55 mL/min/1.73 m^2^, randomized to a usual-protein diet or a LPD [4]. The results of this study were somehow disappointing given the small absolute benefit of approximately 1.1 mL/min/year of GFR associated with a LPD as compared to a standard protein diet. A long-term (12 years) follow-up analysis of the MDRD study, revealed a significant benefit of low-protein intake on renal failure (hazard ratios (HRs) 0.68, CI 0.51–0.93) and all-cause mortality (HR 0.66, CI 0.50–0.87) after the first six years [5]. However, there was no benefit of protein restriction when outcomes between 6 and 12 years were analyzed, and this may be due to the fact that study participants were no longer on the intervention.

In addition to these studies, two recent meta-analysis also showed conflicting effects of LPD on CKD progression in diabetic renal disease. The first one, including eleven randomized controlled trial failed to show any improvement of renal function by LPD in either type 1 or 2 diabetic nephropathy [6]. More recently, another meta-analysis of twenty articles with a total of 690 patients in the LPD and a total of 682 patients in the control group, revealed an effective role of LPD in improving diabetic nephropathy [7]. These results have to be regarded with caution since the heterogeneity was really high, presumably related to the type of diabetes, stages of CKD, types of interventions, duration, and adherence to recommendations. 

A particular situation is represented by the CKD patients with nephrotic proteinuria, where the issue of protein restriction is controversial. A low protein diet coupled with reduced sodium intake may enhance the effects of angiotensin-modulation therapy in decreasing intraglomerular pressure and may also decrease proteinuria and slow the progression of kidney disease. However, concern exists that protein-restricted diets may increase the risk of protein malnutrition. Therefore, most nephrologists recommend no restrictions or only mild restriction in protein intake (0.8–1 g/kg daily), preferring more safer methods such as ACE inhibitors in order to decrease intraglomerular pressure in CKD [4,8]. 

**Table 1 nutrients-13-00956-t001:** Studies of protein restriction alone in chronic kidney disease (CKD) patients.

Name of the Study/Type/Duration/Sample Size	Type of Intervention	Results
**Blood pressure**		
Hansen et al., 2002 [9]/RCT/*n* = 72/ Stage 1, 2, and 3 CKD patients	LPD (38)—0.6 g/kg/d vs. Usual Protein diet group (*n* = 34)	Blood pressure changes were comparable in the two groups during follow-up period.
Meloni, 2002 [10]/RCT/*n* = 69 stage 3, CKD	Normal Protein Diet (12 months) vs. LPD (12 months)—0.6 g protein/kg body weight/day	No differences in blood pressure between the groups
**CKD Progression**		
D’Amico et al., 1994 [11]/RCT/ *n* = 128 Stage 5 CKD patients	LPD—0.6 g/kg vs. Normal protein diet	Normal protein was associated with higher risk of progression compared to LPD
Cianciaruso et al., 2009 [12]/RCT/32 months/*n* = 423 stages 4 and 5 CKD	LPD: 0.55 g/kg/d vs. MPD: 0.8 g/kg/d	No effect of diet assignments was noted on eGFR and proteinuria.
Hansen et al., 2002 [9]/RCT/ *n* = 82 Stage 1, 2, and 3 CKD patients	LPD—0.6 g/kg/d vs. Usual Protein diet group	The difference between group was insignificant
Locatelli et al. [13]/RCT/2 years/ *n* = 456 Stage 3 CKD	LPD—0.6 g/kg vs. Normal protein diet group	No significant difference between the diet groups in cumulative renal survival
Meloni, 2002 [10]/RCT/12 months/ *n* = 69 Stage 3 CKD	LPD 0.6 g protein/kg body weight/day vs. Normal protein diet	The decline in GFR during the study duration was not significantly different between the 2 groups
Rosman et al., 1989 [14]/RCT/18 months/*n* = 207 patients with creatinine clearance ranging from 10 to 60 mL/min	LPD 0.4–0.6 g/kg/d protein intake vs. standard management	Patients who had primary glomerular disease responded very well to the diet and not much effect was seen in others patients.
Sanchez et al., 2010 [15]/RCT *n* = 64 stages 3, 4, and 5 patients	LPD—0.6 g protein/Kg body weight/day vs. Controlled protein diet	GFR rates decreased by 17.2% in the control group compared to only 6.9% in low protein group (NS).
Rosman et al., 1985 [16]/RCT *n* = 199 of various stages of CKD	LPD—0.4 to 0.6 g protein/kg/d protein intake vs. CPD	Median serum creatinine concentration significantly increased in the control group
Williams et al., 1991 [17]/*n* = 95 Predialysis	LPD—0.6 g/kg/day vs. CPD—0.8 g/kg/day	No significant difference in mean rate of fall of creatinine clearance
Cianciaruso et al., 2009 [12]/*n* = 423 stages 4 and 5 CKD	LPD (*n* = 200): 0.55 g/kg/d vs. MPD (*n* = 192): 0.8 g/kg/d	Both groups maintained body weight and 24-hour urinary creatinine excretion similar to the basal value during the entire observation period.
Hansen et al., 2002 [9]/RCT/ *n* = 82 Stage 1, 2, and 3 CKD patients	LPD group: 0.6 g/kg/d vs. CPD	ESRD or death occurred in 27% of Usual protein diet group compared to LPD group (10%) (*p* = 0.042).
**Hard End Points**		
Locatelli et al. [13]/RCT/2 y *n* = 456	LPD 0.6 g/kg/d vs. CPD	The difference between the diet groups in cumulative renal survival was of borderline significance
Rosman et al., 1989 [14]/RCT/18-mo follow-up *n* = 207 patients with creatinine clearance ranging from 10 to 60 mL/min	LPD 0.4–0.6 g/kg/d vs. CPD	Among subjects with low initial creatinine clearances, survival rates were significantly different and in favor of LPD group compared to those in control group (*p* < 0.025).
Rosman et al., 1985 [16]/RCT/*n* = 199 of various stages of CKD	LPD 0.4–0.6 g/kg/d protein intake vs. CPD	Better survival rates for patients on protein restricted diets. People consuming 0.6 g/kg/d of protein had better survival (55%) compared to patients consuming 0.4 g/kg/d of protein (40%).
Cianciaruso et al., 2009 [12]/ RCT/32 months/*n* = 423 stages 4 and 5 CKD	LPD—0.55 g/kg/d vs. MPD—0.8 g/kg/d	Cumulative incidences of death and dialysis therapy start were unaffected by the diet regimen.

LPD—low protein diet, MPD—Moderate protein diet, RCT—randomized controlled trial, CKD—chronic kidney disease, CPD—controlled protein diet, ESRD—end-stage renal disease, eGFR—estimated glomerular filtration rate.

## 3. Protein Restriction and Keto-Analogues

Keto-analogues of amino acids (KAs) are nitrogen-free analogs of essential amino acids. Usually, in combination with either LPD (0.6–0.8 g/kg per day) or very-low-protein diets (VLPD) (0.3–0.4 g/kg per day), they allow a reduced intake of nitrogen while avoiding the deleterious consequences of inadequate dietary protein intake and malnourishment [18]. 

An important limitation of previous trials of protein restriction is that dietary trials have largely focused on restricting total protein rather than on the type of protein intake (animal compared with vegetable). Protein type may be more important for kidney disease progression than the total amount of protein intake, since the increasing intake of red and processed meat is associated with a significantly risk of GFR decline while a strong adherence to a diet characterized by high intake of fruits, vegetables, and low-fat dairy products is associated with a lower risk of CKD [19]. The positive role of a very low protein diet (0.3 g/kg/day) of vegetal origin supplemented with KAs versus a standard low protein diet (0.6 g/kg/day) was highlighted by a randomized controlled trial of 207 patients with a stable eGFR <30 mL/min/1.73 m^2^. After 18 months of follow-up, significantly fewer patients from the KAs group reached the composite endpoint of >50 percent reduction in eGFR or initiation of renal replacement therapy as compared to the low protein diet group (RRT; 42 versus 13 percent, respectively) [20].

Moreover, supplementation of a LPD/VLPD with KAs seems to have some advantages beyond kidney outcomes including preserved eGFR and declined proteinuria. Thus, in a recently published meta-analysis of seventeen RCTs with 1459 participants, KAs appears to provide more effectiveness in lowering blood pressure, nutritional outcomes including increased serum albumin and decreased serum cholesterol, and CKD-MBD parameters comprising diminished serum phosphate and reduced PTH level [21]. These occurred without disturbances in nutritional and anemia status. 

The most recent nutrition guidelines published in 2020 by the The National Kidney Foundation’s Kidney Disease Outcomes Quality Initiative (KDOQI) recommend a LPD providing 0.55–0.60 g dietary protein/kg body weight/day, or a VLPD providing 0.28–0.43 g dietary protein/kg body weight/day with additional keto acid/amino acid analogs in CKD 3–5 who are metabolically stable to reduce risk for end-stage kidney disease (ESKD)/death (1 A) [2]. In the adult with CKD 3–5 and who has diabetes, the same guideline suggests a dietary protein intake of 0.6–0.8 g/kg body weight per day to maintain a stable nutritional status and optimize glycemic control, but this statement is not graded, being only an opinion of the work group [2]. Although the vegetable protein diets may have beneficial effects on health, the type of protein intake (plant vs. animal) is not specified in the recommendations due to the insufficient evidence in terms of the effects on nutritional status, calcium or phosphorus levels, or the blood lipid profile. Even if evidence and guidelines point out several benefits associated with VLPD supplemented with KAs, some patients may find it difficult to adapt their lifestyle to this diet and maintain it on a long-term basis. The MDRD study showed that only 60% of the subjects were adherent to the prescribed dietary protein intake, reason why some clinicians remain reluctant in prescribing these diets. 

Therefore it is of great importance to educate patients about the importance of dietary therapy with LPD/VLPD for the treatment of CKD and to supervise its inclusion in their eating habits. In clinical practice, the compliance with nutritional therapy is indirectly evaluated by dietary self-reporting questionnaires and interviews. Some other biologic like blood urea nitrogen, serum phosphate levels, or and daily urinary excretion of nitrogen are also indirect indicators of protein intake. Adherence to the prescriptions is linked to clinical conditions, sociodemographic factors, the educational level as well as psychological factors. Strategies to improve adherence for low protein diets include identifying and selecting the appropriate CKD candidates and intensive dietary counselling. An alternate graduate approach might be represented by the progressively reduction of the prescribed protein intake while maintaining an adequate energy status since undernutrition exacerbates the risk for malnutrition and wasting (Table 2).

## 4. Nutritional Aspects in Patients with Advanced CKD and in Transition to Dialysis

Classically, the dietary measures in patients with advanced CKD mainly include a reduced sodium, phosphorus, and protein intake aiming to delay or postpone the onset of uremic toxicity that potentially may have cardio-vascular and renal toxic effects. The recommendation of a low-protein diet remains valid even for advanced stages of CKD aiming to enhance the conservative management of non-dialysis CKD patients especially those with contraindications to dialysis or those who chose to avoid or defer the renal replacement therapy [29]. However, several factors put advanced CKD patients at increased risk for developing protein energy wasting. As CKD advances, there is a constant loss of appetite and food intake, mainly related to increased level of inflammatory cytokines which are further stimulated by uremia disturbed gut microbiota and the pathobacterial overgrowth. Furthermore, there is increased hypercatabolism and net muscle protein degradation with worsening of kidney function, metabolic acidosis, and insulin resistance. Therefore, in order to avoid protein energy wasting, it is important that the dietary energy intake should be maintain at approximately 30 to 35 kcal/kg/day, regardless of the recommended protein intake. 

For dialysis patients, evidence from observational studies indicates that LPD is associated with higher hospitalization rates and higher risk for mortality [30,31]. Thus, there is a unanimity among all existing guideline that a low protein diet should be avoided in order to deal with their higher protein catabolism rate [2,32]. There is however a tendency in some dialysis centers towards a more individualized approach combining a LPD, along with a less-frequent dialysis schedule, with main the goal being of increasing the quality of life [33]. Given the fact that there are no randomized controlled trial to show any significant advantage of incremental dialysis over thrice-weekly dialysis schedule in incident dialysis patients, the decision of stating starting dialysis on a twice-weekly or even once-weekly basis must to be reserved to selected patients with preserved residual renal function, with no clinically significant heart failure and good nutritional status.

## 5. Different Types of Diet in CKD Patients

Plant-based diets have been used with rising popularity in CKD. Plant base diet practically include all eating configurations with a great proportion of plant foods (fruits, vegetables, nuts, seeds, oils, whole grain, legumes, and beans) and may or may not include minor/modest quantities of meat, fish, seafood, eggs, or diary. A vegan diet, the Mediterranean diet, the Dietary Approaches to Stop Hypertension (DASH) diet or healthy eating diet are all in line with the concept of plant based diet.

DASH diet is a holistic eating pattern including legumes, whole grains vegetables, fruits, low-fat dairy products; moderate amounts of poultry, fish, and nuts; and low amounts of sodium, red/processed meats, and alcoholic and sweetened beverages. 

The traditional Mediterranean dietary pattern include a high ingesting of fruits, vegetables, bread and whole-grain cereals, potatoes, beans, nuts and seeds and extra virgin olive oil; low-to-moderate amounts of dairy products, fish and poultry and wine, during meals; sweets only occasionally and red meat less often, at special occasion. 

Healthy dietary patterns were commonly higher in fruit and vegetables, fish, legumes, cereals, whole grains, and fiber, and lower in red meat, salt, and refined sugars. 

These dietary patterns are linked with reduced cardiovascular events and death in healthy adults and those at high risk of cardiovascular disease. The impact of these patterns in CKD patients is still uncertain.

### 5.1. Dietary Pattern and Incident CKD

Healthy dietary patterns were connected with the primary prevention of numerous major health conditions, including type 2 diabetes [34], hypertension, and metabolic syndrome [35]; nevertheless, it is unclear whether a healthy dietary pattern may prevent CKD [36]. 

A recent systematic review and meta-analysis reported an association between a healthy dietary pattern and incident CKD [36]. Eighteen prospective cohort studies including 630,108 adults with a mean follow-up of 10.4 ± 7.4 years were eligible for analysis. With moderate certainty of evidence, the primary analysis established that adherence to a healthy dietary pattern was associated with a lower incidence of CKD (odds ratio (OR) 0.70 (95% CI, 0.60 to 0.82)), and incidence of albuminuria (OR 0.77, (95% CI, 0.59 to 0.99)). Still, it remains imprecise what type of “healthy” dietary pattern may be more helpful, given the high heterogeneity across the studies. In analyses of types of healthy dietary patterns, the authors found that the Mediterranean pattern was inversely associated with CKD, but the DASH score was not consistently associated with incident CKD [37]. 

The relationship between DASH diet and the risk of incident CKD was evaluated in several studies, with varying findings. These studies are presented in Table 3. In a study of nearly 15,000 Atherosclerosis Risk in Communities participants, Rebholz et al. noticed a 16% higher risk of incident CKD for the patients with poor accordance to a DASH diet over a median follow-up of 23 years. Higher red and processed meat intake was associated with high risk for kidney disease, while consumption of other sources of protein, including nuts, legumes, and low fat dairy products, was linked with lower risk for kidney disease [38]. Similarly, the Tehran Lipid and Glucose Study, indicated a significant decreases in the risk of incident CKD associated with greater DASH diet adherence. 

A recent systematic review and meta-analysis found a significant inverse association between DASH dietary patterns and the risk of developing CKD [39]. Stratified analysis showed a marginally significant relationship between DASH dietary patterns and risk of CKD in prospective cohort studies (Pooled risk estimate: 0.79, 95% CI 0.61–1.01; *p* = 0.05), and no significant association in cross-sectional studies. Moreover, an important association was established between DASH dietary patterns and risk of CKD in the studies extracted DASH centered on nutrients (Pooled risk estimate: 0.78, 95% CI 0.63–0.97; *p* = 0.02), compared to the studies extracted DASH based on food groups (Pooled risk estimate: 0.66, 95% CI 0.28–1.58; *p* = 0.35).

**Table 3 nutrients-13-00956-t003:** Studies evaluating Dietary Approaches to Stop Hypertension (DASH) diet and the risk of CKD.

Study	Type/Follow-Up	DASH Diet		Outcome
Smyth et al. [40] (2016)	Prospective (14.3 years)	7 food group +Na: total vegetables, total fruit, whole grains, dairy, sugar-sweetened beverages, nuts and legumes, red/processed meat, and sodium	FFQ: Self-report	HR: 0.85 (0.77–0.94)
Liu et al. [41] (2016)	Prospective (5 years)	9 nutrients: total fat, saturated fat, protein, fiber, cholesterol, calcium, magnesium, and potassium, sodium.	24 h recall: Interview	RR: 0.67 (0.38–1.19)
Rebholz et al. [38] (2016)	Prospective (23 years)	9 nutrients: protein, fiber, magnesium, calcium, potassium, saturated fat, total fat, cholesterol, and sodium	FFQ: Interview	HR: 0.86 (0.78–0.93)
Asghari et al. [42] (2017)	Prospective (6.1 years)	7 food groups+ Na: fruits, vegetables, whole grains, nuts and legumes, low-fat dairy, red processed meats, sweetened beverages and sodium	FFQ: Interview	OR: 0.41 (0.24–0.70)
Azizi et al. [43] (2018)	Prospective 3 years	8 foods and nutrients	FFQ: Interview	OR: 0.58 (0.36–0.92) for subjects with dysglycemia, OR = 0.64 (95% CI 0.48–0.87) subjects dyslipidemia OR 0.62 (95% CI 0.44–0.87) for subjects with high BP
Crews et al. [44] (2014)	Cross-sectional	9 nutrients: protein, total fat, saturated fat, cholesterol, fiber, magnesium, calcium, potassium, and sodium	24 h recall: Interview	OR: 0.31 (0.15–0.66) for Subgroup with poverty OR: 1.36 (0.69–2.70) for Subgroup without poverty
Lee et al. [45] (2016)	Cross-sectional	DASH-US score (based on the US recommendations) DASH-KQ model 6 nutrients: protein, fiber, calcium, potassium, total fat and sodium	24 h recall: Interview	OR: 0.78 (0.65–0.94) DASH-US score (based on the US recommendations) OR: 0.95 (0.91–0.99) DASH-KQ model

FFQ—food frequency questionnaire, HR—hazard ratio, RR—relative risk, OR—odds ratio, CI—confidence interval, Na—sodium.

Superior adherence to the Mediterranean pattern could be also helpful on the prevention of CKD, but the studies in this regard are limited and more analyses to confirm these findings are necessary [46] (see Table 4). In a six-year follow-up study the authors found that participants with higher rates of MD adherence had a 50% lower risk of developing CKD [47]. Data from the Northern Manhattan Study presented the same significant association between MD adherence and CKD prevention. In contrast, in a secondary analysis of data from the National Nutrition and Physical Activity Survey (NNPAS), the largest health study in Australia and the first nutrition-specific national-based study, no significant association was observed among MD adherence and CKD [48]. A recent systematic review and meta-analysis, including four studies (*n* = 8467), was performed. Adherence to Mediterranean pattern by a one-point increment of MDS was associated with 10% lower risk of CKD.

In a prospective study, adherence to modified Alternate Healthy Eating Index (close to DASH diet) was associated with 20% reduction in the incidence or progression of CKD between patients with diabetes after five years of follow-up [56]. Other prospective studies investigated associations between other types of DIET (American Heart Association’s (AHA) Life’s Simple Seven Healthy Diet Score the Dietary Guidelines Adherence Index (DGAI) and the Total Diet score (TDS) and risk of CKD with inconstant results [57]. In the Framingham Offspring Cohort study, greater adherence to the DGAI was associated with a significant decreased risk of CKD. No significant associations for the AHA Healthy Diet Score, or the TDS, were reported [40].

### 5.2. Diet Patterns and CKD Progression 

DASH diet could also be protective for CKD progression. In a cross-sectional analysis of 2408 community-dwelling elderly participants from the Korean National Health and Nutrition Examination Survey, a high DASH score was related with low odds for CKD [58]. The same favorable results were also noted for high fiber intake. In the Nurses’ Health Study, those with better accordance to a DASH diet had lower risk of rapid estimated glomerular filtration rate decline [57]. Similar results are reported from the US, using 1110 adults with hypertension and CKD stage 3 enrolled in the National Health and Nutrition Examination Survey (NHANES) III. A DASH diet accordance score was calculated based on self-reported intake of nine nutrients. Matched with those with an accordance score in the highest quintile, those in the two lowest quintiles had approximately twofold greater risk of progressing to ESKD. These were predominantly strong among those with diabetes. The individual components of the DASH accordance score could be also important. The dietary K and Mg seem to be strong mediators of the association between DASH accordance score and risk of ESRD. This observation suggests that potassium-lite versions of the DASH diet, which might be attractive in this category of patients, could be ineffective [59]. 

A recent cross-sectional German study used a food frequency questionnaire (FFQ) in CKD patients and found that the Mediterranean pattern was associated with better GFR levels even after accounting for all possible confounders [59]. In contrast, an analysis in the PREDIMED Study did not show important differences in kidney function between Mediterranean-style diets compared with control low-fat diet after one year of follow-up [54].

In a recent prospective cohort study including 2403 individuals with CKD from the Chronic Renal Insufficiency Cohort (CRIC) cohort, several diet scores were calculated from food frequency questionnaires; higher diet quality was consistently associated with lower risk of progression. The association between aMed and CKD progression was the strongest. The strong association in aMed may be due to the individual components included in the score.

A systematic review that 1639 people with CKD did not find healthy dietary interventions (e.g., DASH diet, Mediterranean diet, American Heart Association diet) to influence ESRD, but were related with lower blood pressures and low-density lipoprotein cholesterol. Nevertheless, the quality of evidence was very low for ESRD (short follow-up; limited number of events) [60]. 

In the prospective observational Northern Manhattan Study (*n* = 900), plant-based diets were linked with a 12% lower risk of decline in eGFR compared with meat-based diets [52]. A “vegan” diet could also be favorable. In a study of elderly, nondiabetic patients, with no symptomatic uremia, dialysis was delayed for about a year in the “vegan” group. This was achieved without increasing the risk of death or hospitalization [61].

### 5.3. Dietary Patterns and Mortality

The relationship between healthy dietary scores and all-cause mortality was also investigated. Healthy Eating Index-2015, Alternative Healthy Eating Index-2010, alternate Mediterranean diet (aMed), and Dietary Approaches to Stop Hypertension (DASH) diet scores were calculated from food frequency questionnaires. Higher diet quality based on each of the four dietary patterns evaluated was constantly related with lower risk of death.

Similar results were provided by a recent meta-analysis of seven cohort studies, including 15,285 participants; it showed that a healthy dietary patterns were consistently associated with lower mortality (adjusted relative risk: 0.73, 95% CI: 0.63–0.83). A previous (2017) review that incorporated 17 randomized clinical trials of 1639 people with CKD did not find a promising effect of the food-based dietary interventions (e.g., DASH diet, Mediterranean diet, American Heart Association diet on ESRD, CVD, or all-cause mortality [60]. 

An appropriate nutrition represents an essential part in offering care for dialysis patients, not only for a better quality of life but also for a lower mortality risk. However, recent data found no associations between vegetarian diets and total and CVD mortality. In a recent study involving 8110 patients on HD from ten European countries and one South American country, there was no evidence of an association between Mediterranean and DASH diets and cardiovascular or all-cause mortality in patients on hemodialysis. Excluding participants who died in the first 12 months of follow-up did not alter the association between diets scores and cardiovascular or all-cause mortality. Age was a significant effect modifier of the relationship between DASH diet score and all-cause mortality but not cardiovascular mortality. 

Interestingly, a higher number of fruits and vegetables consumed per week was associated with lower risk of all cause and non-cardiovascular mortality in hemodialysis patients [62]. It is well known fruits and vegetable intake are considerably lower in those on hemodialysis. However, in comparison to two servings of fruits and vegetables per week, 17 servings per week are associated with a 20% lower hazard of all-cause mortality (HR, 0.80; 95% CI, 0.71 to 0.91) and a 23% lower hazard of non-cardiovascular mortality (HR, 0.77; 95% CI, 0.66 to 0.91). It is documented that dialysis patient education tools frequently incorporate food lists; those with “high potassium" are avoided, although data in a recent study found no significant correlations between reported mean potassium consumption and pre-dialysis serum potassium level. 

In kidney transplant patients, the adherence to a Mediterranean pattern was associated with a lower incidence of the metabolic syndrome [63]. Similarly, adherence to the DASH diet was related with a lower risk of kidney function decline and a lower risk of death during a median follow-up of 5.3 years [64]. 

## 6. Possible Mechanisms

Several possible mechanisms can be speculated:

(1) Diminished acid load—the incapacity of the kidney to excrete the daily dietary load acid predispose to the development of overt metabolic acidosis. It has already been shown that metabolic acidosis is associated with impaired insulin secretion, increased risk of hypertension, heart failure, muscle wasting, chronic inflammation, and progression of CKD and increased mortality [65]. Metabolic acidosis increases the renal production of endothelin and angiotensin II, with the succeeding stimulation of the aldosterone and an increased in renal mediated ammoniogenesis, causing kidney damage. Animal protein is acid forming due to the existence of organic sulfur, which is found in the amino acids methionine and cysteine (common in animal proteins) and is reacted to inorganic sulfate on the other hand; plant-based foods have natural dietary alkali, in the form of citrate and malate, which can be converted to bicarbonate [66,67]. 

(2) Reduction in serum phosphate and a decreased urinary 24 h phosphorus excretion—The absorption of the phosphorus from plant sources is decreased (less than 30–40%) compared with phosphorus from animal sources (80%), due to the presence of phytate in plants, which causes phosphorus less bio accessible for gastrointestinal absorption. Additionally, animal-based food contains food additives, who can increase the amount of bioavailable phosphate by 7–100% [68]. Small studies indicated significant reduction in serum phosphorus and FGF 23 levels and a reduced urinary 24-h phosphorus excretion in the vegetarian group compared with animal-based group [69]. 

(3) Reduction of metabolites resulting from gut bacteria that are linked with CKD and CV disease—It has already been presented that in CKD patients the gut microbiota is characterized by a disproportion between commensal bacteria, which are reduced and uremic toxins producing bacteria and pathobionts, which are augmented. The key modifiers of gut microbiota are the dietary habits [70]. Animal protein ingredients containing choline and carnitine are converted by gut flora into trimethylamine (TMA) and TMA N-oxide (TMAO) that are associated with atherosclerosis, renal fibrosis, and expanded risk of CV disease and death; a plant based diet is a fiber rich diet; it was already demonstrated the following favorable effects of higher fiber intake: (a) it stimulates fermentation and the saccharolitic catabolism of food, causing in a downstream increase in beneficial short chain fatty acids (butyrate, propionate or acetate) production; as a consequences the insulin sensitivity increases and the serum cholesterol and glucose absorption decreases. It also traps protein nitrogen, resulting in amplified nitrogen excretion; (b) decreases the amount of time for protein fermentation in the intestinal tract; in this context, several bacterial metabolites such as ammonia, phenols, indols, etc., are decreased; and (c) by increasing intestinal motility, the time for fermentation of amino acids declines, and augments the elimination of human or bacterial products, decreasing the formation and the accumulation of the uremic toxins. 

(4) Anti-inflammatory and antioxidant effects: the dysbiotic gut microbiome augments pathobionts and increases intestinal barriers, promoting systemic inflammation and oxidative stress in CKD patients through the translocation of bacteria and bacterial products into the systemic circulation [71]. Consumption of a Mediterranean-style diet was previously associated with improvement of endothelial function and a significant reduction of markers of systemic vascular inflammation. Further, plasma markers of chronic inflammation were significantly reduced in CKD patients during the combined consumption of white wine and olive oil [72]. 

There are numerous major concerns about these types of healthy diets in CKD patients. The most significant is clearly hyperkalemia. Recent data delivered by observational study do not establish an undoubted link between dietary potassium and circulating potassium levels in people with CKD, comprising those on hemodialysis [73]. An imprecision in the calculation of potassium in the diet could be a possible explanation. The diverse sources of potassium with different bioavailability, the use of potassium additives or potassium salt substitution or underestimation of the quantity of potassium that is lost in cooking can determine this erroneousness. Furthermore, drugs (such as such as renin–angiotensin–aldosterone system inhibitors and comorbidities (such as heart failure, diabetes, or metabolic acidosis) could influence the serum level of potassium. Moreover, evidence showing dietary potassium restriction causes a reduction in serum potassium, provided by RCTs appears limited. Very-low-quality evidence provided by a systematic review and meta-analysis of 2 randomized trials and 5 observational studies involving 3489 normokalemic patients with CKD stages 3 to 5 D supports consensus that dietary potassium restriction reduces serum potassium and is associated with a reduced risk of death in those with CKD. In contrast, among 219 non-dialysis CKD patients managed in a renal nutrition clinic, plant-based low-protein diet is not associated with significant higher prevalence of HK with respect to animal-based LPD at the same residual kidney function. Hyperkalemia was not connected with higher risk of mortality, but a trend, although not statistically significant, was detected for lower ESRD-free survival. In a study of 43,798 mostly male US veterans transitioning to end-stage kidney disease, high-normal serum potassium levels of at least 5.0 but less than 5.5 mEq/L were associated with 5% greater survival after dialysis initiation compared with a reference of at least 4.5 but less than 5.0 mEq/L. In contrast, low-normal serum potassium concentrations of at least 4.0 but less than 4.5 mEq/L and at least 3.5 but less than 4.0 mEq/L and below-normal levels of less than 3.5 mEq/L were associated with 7%, 5%, and 16% higher risks for all-cause mortality, respectively. 

## 7. Conclusions

Healthy dietary patterns higher in fruit and vegetables intake and lower in protein intake of animal origin may have potential benefits in slowing CKD progression, in postponing the onset of uremic toxicity in advanced stages of CKD and even in decreasing mortality. Protein-restricted diets have other important metabolic benefits including decreasing sodium intake and improving blood pressure control in patients with advanced CKD [22]. However, the long-term effects of dietary patterns on health outcomes are difficult to separate from lifestyle elements associated with a vegetarian diet (e.g., being physically active, avoidance of tobacco, and moderate alcohol products).

These potential benefits of these diets are sometimes counterbalanced by a poor patients’ adherence to these prescription. Thus, a personalized approach is mandatory in identifying and selecting the appropriate CKD candidates for nutritional therapy since some patients may prefer to enjoy their food without restriction, despite knowing the potential importance of dietary therapy in reducing uremic manifestations. Patients’ preference and compliance have to be considered when prescribing a low protein diet in advanced CKD patients in order to increase adherence. Therefore, in-depth education, nutritional education programs, and regular close monitoring by a renal dietician is mandatory for implementation of a low, or very low, protein diet in order to increase the adherence, and thus protect the kidneys. 

## Figures and Tables

**Table 2 nutrients-13-00956-t002:** Studies of protein restriction +KAs in CKD patients.

Name of the Study/Type/Duration/Sample Size	Type of Intervention	Results
Bellizi et al., 2007 [22]/Non RCT/*n* = 114 Stages 4 and 5 CKD	VLDL 0.3 g protein/kg/day + mixtures of KAs & EAA- mean total protein intake of 0.35 g/kg/d vs. LPD group: *n* = 57; 0.6 g proteins/kg/day	VLPD patients showed a significant reduction in SBP and DBP and more patients reached the BP target. Proteinuria significantly decreased only in VLPD group
Garneata et al., 2016 [20]/RCT/*n* = 207 stage 4 and 5 CKD	VLPD group (*n* = 104): Protein intake 0.3 g/Kg/d + KAs vs. LPD group (*n* = 103): Protein intake 0.6 g/kg/d	BP was controlled with antihypertensive medications. The decrease in eGFR was lower in KD compared with LPD Significantly lower patients in the VLPD + KAs group reached the primary end point (>50% reduction in GFR or RRT) compared to LPD group
Herselma et al., 1996 [23]/RCT/*n* = 22 predialysis patients	LPD group (*n* = 11)—0.6 g/kg/day vs. VLPD + EAA group (*n* = 11): 0.54 g/kg/d (0.4 g/kg + 0.14 g EAA/kg)	No effect of smented VLPD on BP, protein-energy status, renal function and biochemical parameters as compared to the LPD group No difference between the groups regarding rate of progression before or during the study.
Mircescu et al., 2007 [24]/RCT/48 weeks/*n* = 53 Stages 4 and 5	VLPD (*n* = 27): 0.3 g/kg/day (vegetable proteins) + mixture of EAA vs. LPD (*n* = 26): 0.6 g/kg/day	Estimated GFR did not change significantly in patients receiving VLPD + KAs but significantly decreased in the LPD group A significantly lower percentage of patients in the VLPD + KAs group required RRT initiation throughout the therapeutic intervention
Feiten et al., 2005 [25]/RCT/4 months/*n* = 24 Stage 4 pa-tients	VLPD + KAs (*n* = 12): 0.3 g/kg/day of vegetal origin protein diet + KAs vs. LPD (*n* = 12): 0.6 g/kg/day of protein	No difference between the two groups regarding the creatinine and the creatinine clearance
Klahr 1994 [4]/RCT/*n* = 840 Stage 3 and 4 CKD	Study 1 (patients with GFR= 25–55 mL/min/1.73 m^2^) LPD vs. usual PD Study 2 (patients with GFR = 13–24 mL/min/1.73 m^2^)VLPD + KAs vs. usual PD	Study 1—the mean rate of GFR decline did not differ significantly between the 2 groups Study 2—trend for slower GFR decline in the very low-protein group when compared with the low-protein group
Prakesh et al., 2004 [26]/RCT/9 months/*n* = 18 Stage 4	Keto-diet group (*n* = 18): 0.3 g/kg protein + KAs Placebo group (*n* = 16): 0.6 g/kg/d protein + placebo tablets	GFR stayed unchanged in the Keto-acid group, however, it significantly decreased in the placebo group.
Levey, 1996 [27]/ RCT/2.2 years (0–44 mo)/*n* = 255 Predialysis Stages 3 and 4	LPD: 0.58 g/kg/d vs. VLPD: 0.28 g/kg/day smented with keto acids–amino acids	Assignment to the very low-protein diet did not have a significant effect on renal failure/death risk. Lower protein intake, but not the keto acid–amino acid sment, retards the progression of advanced renal disease
Malvy et al., 1999 [28] *n* = 50/Stages 4 and 5	VLPD: 0.3 g/kg/d + 0.17 g/kg/day ketoanalogues vs. LPD: 0.65 g/kg/d protein intake	No difference between the two groups when comparing renal survival

RCT—randomized controlled trial, KAs—ketoanalogues, VLPD—very low protein diet, LPD—low protein diet, EAA—essential amino-acids, SBP—systolic blood pressure, DBP—diastolic blood pressure, GFR—glomerular filtration rate, RRT—renal replacement therapy.

**Table 4 nutrients-13-00956-t004:** Studies evaluating Mediterranean pattern and the risk of CKD.

Study	Type/Follow-Up	*n*	Mediterranean Diet	Outcome
	NNPAS	9435	Mediterranean diet adherence score;	CKD: 0.99 (0.9–1.06)
Hu et al., 2019 [49]	Prospective 24 years	5245	Mediterranean diet scale;	CKD: 0.91 (0.87–0.95)
Heindel et al. [50], 2020	Prospective 2 years	2813	Mediterranean diet scale;	eGFR: β-coefficient = 0.932, *p* = 0.007
Asghari et al. [47], 2017	Prospective 6 years	1212	Mediterranean diet scale;	OR = 0.53; 95% CI: 0.31–0.91
Alkerwi et al. [51], 2015	Prospective 6.1 years	1352	Mediterranean diet scale;	SCr β −0.884, *p* 0.004
Khatri et al. [52],2014	Prospective 6.9 years	900	Mediterranean diet scale;	OR, 0.83; 95% CI, 0.71 to 0.96
Huang et al. [53], 2013	Prospective 9.9 years	1110	Mediterranean diet scale;	CKD—medium adherence—OR = 0.77 (0.57 to 1.05) high adherents OR 0.58 (0.38 to 0.87), *p* for trend = 0.04
Diaz-Lopez et al. [54], 2012	Prospective 1 years	785	Mediterranean diet adherence score;	eGFR + 4.7 (95% CI, 3.2–6.2) − MedDiet + olive oil, + 3.5 (95% CI, 1.9–5.0 mL/min/1.73 m^2^) MedDiet + nuts
Chrysohoou et al. [55], 2010	Prospective 5 years		Greek European Prospective Investigation into Cancer and Nutrition (EPIC) food frequency questionnaire	Multivariate linear regression: CCR β 0.003, *p* 0.06

National Nutrition and Physical Activity Survey (NNPAS); MDS—Mediterranean diet scale, eGFR—estimated glomerular filtration rate, OR—odds ratio, CI—confidence interval, CCR—Correlated Component Regression

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
