# Peer review of "Nutrition in Chronic Kidney Disease—The Role of Proteins and Specific Diets"

_nutrients, 2021, doi:10.3390/nu13030956_

Round 1

Reviewer 1 Report

This is a well-written review updating a highly challenging topic within the nutrition-related questions that arise for patients with chronic kidney disease – how should protein intake be approached. The authors provide a balanced and accurate scientific tone and a well-documented presentation of this issue. However, tackling the following points may bring additional clarity to the submitted manuscript:

  1. I would suggest the authors create an Introduction section from the first paragraphs of the paper and delineate the last paragraph in the text within a Conclusions section.
  2. What is the current status of clinical evidence concerning long-term patients’ adherence to low-protein diets? The authors shortly mention this issue, but without detailing it. As the adherence topic is considered a high-profile approach within the modern management of patients with any chronic disease, I think the addition of a short fragment on this theme might increase the utility of this review paper for a larger number of researchers, and the citation interest accordingly.
  3. The quality of the English language is generally good. However, I would advise the authors to reconsider the following constructs:

“The role of nutritional therapy [...] have been recommended” (lines 8-9)

“will further impair organ, function” (lines 36-37)

“a high protein is responsible” (line 37)

“being related type of diabetes” (line 81)

“LPD intake” (line 92) (the “diet intake” apposition is unnecessary)

“very low protein diet plant-based protein diet” (lines 115-116)

“in recent published meta-analysis” (line 124)

“the authors recommends” (line 136)

Author Response

Reviewer’s Critic: I would suggest the authors create an Introduction section from the first paragraphs of the paper and delineate the last paragraph in the text within a Conclusions section. 

Authors’ Reply: We would like to thank you for your comments. We changed significantly the structure of the manuscript according to your suggestion. 

Reviewer’s Critic: What is the current status of clinical evidence concerning long-term patients’ adherence to low-protein diets? The authors shortly mention this issue, but without detailing it. As the adherence topic is considered a high-profile approach within the modern management of patients with any chronic disease, I think the addition of a short fragment on this theme might increase the utility of this review paper for a larger number of researchers, and the citation interest accordingly.

Authors’ Reply: Done. We took into consideration your criticism. Therefore, we included more evidence and more relevant details regarding adherence to low-protein diets

Reviewer’s Critic The quality of the English language is generally good. However, I would advise the authors to reconsider the following constructs:

Authors’ Reply: We made all the suggested corrections.

“The role of nutritional therapy [...] have been recommended” (lines 8-9) Done

“will further impair organ, function” (lines 36-37) Done

“a high protein is responsible” (line 37) Done

“being related type of diabetes” (line 81) Done

“LPD intake” (line 92) (the “diet intake” apposition is unnecessary) Done

“very low protein diet plant-based protein diet” (lines 115-116) Done

“in recent published meta-analysis” (line 124) Done

“the authors recommends” (line 136) Done

Reviewer 2 Report

The manuscript under the title “NUTRITION IN CHRONIC KIDNEY DISEASE- THE ROLE 2 OF PROTEINS AND SPECIFIC DIETS' attempts to do an overview about protein,  nutrient patterns and CKD

According to my point of view it is as if two different reviews have been put together. The introduction includees information only for protein, but not only protein and CKD but basic knowledge on proteins and needs general reconsideration. In Page 2 lines 93-95 you mention that the high protein intakes in hemodialysis are reported in the recent KDOQI guidelines. High protein intake in Hemo and peritoneal dialysis is suggested in ALL the available guidelines for CKD and this part should either omitted or said differently.

You do not include methodology on how you did the research, the MESH terms and an algorithm for the selection of the articles you used.

Mediterranean diet is not a special diet but rather a dietary pattern so I would suggest to change the phrasing throughout the manuscript

The conclusion corresponds only to the diets but not in the protein and it is not sufficient.

There are two sets of References. Please correct the list and provide a revised list as the references do not correspond to the manuscript.

Overall it is not a very well written review, with not clear methods of selection of the papers included, dealing with two  parameters of the management of CKD but in a way as if you try to combine two different reviews. I would suggest to do two different reviews or do an attempt to better homogenize this result.

Author Response

Reviewer’s Critic: According to my point of view it is as if two different reviews have been put together. The introduction includes information only for protein, but not only protein and CKD but basic knowledge on proteins and needs general reconsideration.

Authors’ Reply: We would like to thank the reviewer for this comment. We took into consideration your criticism and re-arranged different parts and general editing of the entire manuscript; furthermore, the connection between sections was overhauled. Some paragraphs and sections that were irrelevant– were deleted.

Reviewer’s Critic In Page 2 lines 93-95 you mention that the high protein intakes in hemodialysis are reported in the recent KDOQI guidelines. High protein intake in Hemo and peritoneal dialysis is suggested in ALL the available guidelines for CKD and this part should either omitted or said differently.

Authors’ Reply: We changed the paragraph according to your observation.

Reviewer’s Critic You do not include methodology on how you did the research, the MESH terms, and an algorithm for the selection of the articles you used.

Authors’ Reply: Thank you again for your useful suggestion. We included a short paragraph with the methodology used for this narrative review.

Reviewer’s Critic Mediterranean diet is not a special diet but rather a dietary pattern so I would suggest changing the phrasing throughout the manuscript

Authors’ Reply: We appreciate the reviewer’s opinion and we decided to replace the term Mediterranean diet with Mediterranean pattern throughout the manuscript

Reviewer’s Critic The conclusion corresponds only to the diets but not in the protein and it is not sufficient.

 Authors’ Reply: Thank you again for your criticism. We changed considerably the entire manuscript including the conclusion section.

Reviewer’s Critic There are two sets of References. Please correct the list and provide a revised list as the references do not correspond to the manuscript.   DONE

Authors’ Reply: Done. We correct the list reducing the number of references and maintaining only those we considered most relevant for the topic

Reviewer’s Critic Overall it is not a very well written review, with not clear methods of selection of the papers included, dealing with two  parameters of the management of CKD but in a way as if you try to combine two different reviews. I would suggest to do two different reviews or do an attempt to better homogenize this result.

Authors’ Reply: We would like to thank you for your comments. We did a thorough analysis of the text focusing on redrawing the connection between several parts of the manuscript according to your suggestion.

Reviewer 3 Report

In their narrative review, Apetrii and colleagues report a good summary of the available evidence on the impact of low-protein diets and dietary patterns on chronic kidney disease-related outcomes. The review is well-written and easy to follow; I have a few minor comments and suggestions that could further improve the manuscript:

  • Indications from KDOQI guidelines are repeatedly reported in several sections of the manuscript (page 2 line 93; page 5 line131; page 7 line 170); please consider condensing them in a single paragraph.
  • I would suggest avoiding bulleted lists (e.g. page 7 lines 193-202 and page 12) to increase readability; in particular, a table summarizing the key features and differences of plant-based diets (Mediterranean, DASH, vegan…) could be included instead.
  • In the paragraph “Nutritional aspects in patients with advanced CKD and in transition to dialysis”, I believe that the authors could briefly discuss the available evidence and the rationale of dietary management and incremental hemodialysis to preserve residual renal function in patients who are transitioning to dialysis.
  • I think that the discussion on theoretical biological mechanisms that may explain the effect of dietary patterns on CKD outcomes (page 12) would deserve a separate chapter.
  • At page 11, lines 323-333: the authors state that “There are 324 some evidence suggesting that an optimal diet could be associated with improved sur-325 vival not only in CKD, but also in dialysis patients”, but the following sentences refer to negative outcomes of the referenced study. Please consider re-organizing this paragraph by highlighting positive results first, in order to support the previous statement.
  • Please include the relative reference at page 12, lines 347-348.
  • Consider correcting typos / revising the following sentences:
    • Page 7, line 182: “Plant-based diets have been used with growing popularity for the treatment of a wide 182 range of lifestyle-related diseases, CKD”.
    • Page 7, line 186: “which is a plant based diet” is redundant
    • Page 8, line 237: “founded”
    • Page 10, line 257: “a dietary pattern rich in fruits and vegetables” is redundant
    • Page 10, line 257: “aged”
    • Page 11, lines 312-314: consider omitting the sentence “Healthy Eating Index-2015, Alternative Healthy Eating Index-2010, alternate Mediterra-312 nean diet (aMed), and Dietary Approaches to Stop Hypertension (DASH) diet scores were 313 calculated from food frequency questionnaires”, since it was already specified in the previous paragraph.

Author Response

Reviewer’s Critic Indications from KDOQI guidelines are repeatedly reported in several sections of the manuscript (page 2 line 93; page 5 line131; page 7 line 170); please consider condensing them in a single paragraph.

Authors’ Reply: Done. We took into consideration your observation and made a single paragraph with the KDOQI nutritional recommendation.

Reviewer’s Critic I would suggest avoiding bulleted lists (e.g. page 7 lines 193-202 and page 12) to increase readability; in particular, a table summarizing the key features and differences of plant-based diets (Mediterranean, DASH, vegan…) could be included instead.

Authors’ Reply. Thank you for your comment. We removed the bullets accordn to your suggestion.

Reviewer’s Critic In the paragraph “Nutritional aspects in patients with advanced CKD and in transition to dialysis”, I believe that the authors could briefly discuss the available evidence and the rationale of dietary management and incremental hemodialysis to preserve residual renal function in patients who are transitioning to dialysis.

I think that the discussion on theoretical biological mechanisms that may explain the effect of dietary patterns on CKD outcomes (page 12) would deserve a separate chapter.

Authors’ Reply: We appreciate the reviewer’s opinion and we decided to add some text to tackle this

Reviewer’s Critic At page 11, lines 323-333: the authors state that “There are 324 some evidence suggesting that an optimal diet could be associated with improved sur-325 vival not only in CKD, but also in dialysis patients”, but the following sentences refer to negative outcomes of the referenced study. Please consider re-organizing this paragraph by highlighting positive results first, in order to support the previous statement

Authors’ Reply: Thank you for your observation. We changed the paragraph according to your suggestion.

  •  

Reviewer’s Critic Please include the relative reference at page 12, lines 347-348. Done

Authors’ Reply: We inserted the requested referinces and made all the suggested corrections

  • Consider correcting typos / revising the following sentences:
    • Page 7, line 182: “Plant-based diets have been used with growing popularity for the treatment of a wide 182 range of lifestyle-related diseases, CKD” – done

 « Plant-based diets have been used with rising popularity in CKD

  • Page 7, line 186: “which is a plant based diet” is redundant - done
  • Page 8, line 237: “founded” done
  • Page 10, line 257: “a dietary pattern rich in fruits and vegetables” is redundant - done
  • Page 10, line 257: “aged” - done

Page 11, lines 312-314: consider omitting the sentence “Healthy Eating Index-2015, Alternative Healthy Eating Index-2010, alternate Mediterra-312 nean diet (aMed), and Dietary Approaches to Stop Hypertension (DASH) diet scores were 313 calculated from food frequency questionnaires”, since it was already specified in the 

Round 2

Reviewer 2 Report

The manuscript has improved significantly but there are still problematic areas regarding the English language use. There are several mistakes in grammar and they should be checked by a native English speaker.